# Aberrant B Cell Receptor Signaling in Naïve B Cells from Patients with Idiopathic Pulmonary Fibrosis

**DOI:** 10.3390/cells10061321

**Published:** 2021-05-26

**Authors:** Stefan F. H. Neys, Peter Heukels, Jennifer A. C. van Hulst, Jasper Rip, Marlies S. Wijsenbeek, Rudi W. Hendriks, Odilia B. J. Corneth

**Affiliations:** Department of Pulmonary Medicine, Erasmus MC, University Medical Center, 3015 GD Rotterdam, The Netherlands; s.neys@erasmusmc.nl (S.F.H.N.); pheukels@amphia.nl (P.H.); j.vanhulst@erasmusmc.nl (J.A.C.v.H.); j.rip@erasmusmc.nl (J.R.); m.wijsenbeek-lourens@erasmusmc.nl (M.S.W.)

**Keywords:** idiopathic pulmonary fibrosis (IPF), B cell receptor (BCR) signaling, nintedanib, Bruton’s tyrosine kinase (BTK), autoimmunity

## Abstract

Idiopathic pulmonary fibrosis (IPF) is a chronic and ultimately fatal disease in which an impaired healing response to recurrent micro-injuries is thought to lead to fibrosis. Recent findings hint at a role for B cells and autoimmunity in IPF pathogenesis. We previously reported that circulating B cells from a fraction of patients, compared with healthy controls, express increased levels of the signaling molecule Bruton’s tyrosine kinase (BTK). However, it remains unclear whether B cell receptor (BCR) signaling is altered in IPF. Here, we show that the response to BCR stimulation is enhanced in peripheral blood B cells from treatment-naïve IPF patients. We observed increased anti-immunoglobulin-induced phosphorylation of BTK and its substrate phospholipase Cγ2 (PLCγ2) in naïve but not in memory B cells of patients with IPF. In naïve B cells of IPF patients enhanced BCR signaling correlated with surface expression of transmembrane activator and calcium-modulator and cyclophilin ligand interactor (TACI) but not B cell activating factor receptor (BAFFR), both of which provide pro-survival signals. Interestingly, treatment of IPF patients with nintedanib, a tyrosine kinase inhibitor with anti-fibrotic and anti-inflammatory activity, induced substantial changes in BCR signaling. These findings support the involvement of B cells in IPF pathogenesis and suggest that targeting BCR signaling has potential value as a treatment option.

## 1. Introduction

Idiopathic pulmonary fibrosis (IPF) is a chronic and irreversible interstitial lung disease (ILD) of unknown cause, characterized by an imaging and pathological pattern of interstitial pneumonia [1]. IPF has been associated with various risk factors, including cigarette smoking, pulmonary microbiome, and genetic predisposition. The disease is thought to develop slowly with a dysregulation in the response to subclinical recurrent pulmonary micro-injuries, leading to abnormal tissue remodeling [2]. IPF has a poor median survival of 3–5 years following diagnosis. Because effective therapies are limited, unraveling of the disease pathogenesis is required to explore novel treatment options that slow disease progression or that may even prevent disease development.

Currently, two treatment options for IPF patients are recommended, nintedanib and pirfenidone, both of which modulate fibrosis. Nintedanib is a tyrosine kinase inhibitor that inhibits signaling downstream of pro-fibrotic receptors, such as fibroblastic and vascular endothelial growth factor receptors (FGFR and VEGFR) [3]. The exact mechanism by which pirfenidone exerts its clinical effect is not completely understood, but it is known to have the capacity to modulate both pro-fibrotic and pro-inflammatory factors, such as transforming growth factor (TGF)-β, tumor necrosis factor (TNF)-α, and interleukin (IL)-6 [4,5]. Although these therapies delay the decrease in lung function and improve survival, better therapies that control disease progression are in high demand.

An increasing body of evidence suggests that B cells and autoimmunity play a role in the pathogenesis of IPF [6]. The lungs of IPF patients often contain lymphocytic aggregates that co-localize with fibrotic areas [7,8,9]. Levels of the B cell chemoattractant C-X-C motif ligand 13 (CXCL13) are increased both locally in the lungs and systemically in the serum of IPF patients compared with healthy controls (HCs), and are predictive of survival [10,11]. Similarly, B cell activating factor (BAFF) and immunoglobulin (Ig)A levels are increased in the serum of IPF patients and are similarly predictive of survival [12,13]. Several studies have shown the presence of autoreactive antibodies of various specificities in the serum of IPF patients, especially during acute exacerbations (AE) of the disease [14]. Concordantly, beneficial effects of B cell-targeting therapies in AE-IPF patients have been suggested, such as plasma exchange in combination with rituximab or intravenous Ig [15,16]. The involvement of B cells in IPF pathogenesis is further supported by the observation of a significant increase in activated follicular T helper (Tfh) cells both in the circulation and locally in the lungs of IPF patients [17,18].

Bruton’s tyrosine kinase (BTK) is best known for its role in signal transduction downstream of the B cell receptor (BCR) [19]. We recently identified increased levels of BTK in naïve B cells as an overarching characteristic of systemic autoimmune diseases, including rheumatoid arthritis (RA), primary Sjögren’s syndrome (pSS), and granulomatosis with polyangiitis (GPA) [20,21]. In addition, we showed that B cell-specific overexpression of human BTK under the control of the CD19 promotor in CD19-hBtk transgenic mice leads to the development of a T cell-dependent systemic autoimmune phenotype [22,23]. Moreover, proteome profiling of IPF peripheral blood indicated highly increased BTK expression compared with HCs [24].

In line with these results, we observed that in a major fraction of IPF patients, BTK protein expression in resting peripheral blood B cells was increased compared with HCs [18]. Given that BTK is also phosphorylated downstream of many other receptors in B cells [19], it remains to be determined whether the increased BTK protein expression in IPF is linked to a general enhancement of BCR signaling. It is then conceivable that BTK activity is not only enhanced in resting B cells, but also upon BCR engagement. This may be a reflection of an altered activity of various upstream molecules, including spleen tyrosine kinase (SYK), which upon activation can subsequently phosphorylate and activate BTK. Moreover, it is unknown whether increased BTK protein expression in resting B cells translates into increased downstream signaling, involving phosphorylation of its primary substrate, phospholipase Cγ2 (PLCγ2), or activation of the phosphoinositide-3-kinase (PI3K)-AKT-mTOR (mammalian target of rapamycin) pathway.

It has been shown that survival of naïve B cells critically depends on low-level—so-called “tonic”—BCR signals in the absence of antigen engagement and pro-survival signals from the BAFF receptor (BAFFR). Hereby, the latter was demonstrated to transduce survival signals by crosstalk with the BCR, involving SYK and BTK [25,26]. It is therefore possible that the increase in BAFF levels in IPF patients is linked to aberrant BTK levels and activity, and hence BCR signaling, in naïve B cells. To address these questions, in this report we aim to investigate BCR signaling, particularly focusing on SYK, BTK, and PLCγ2 activity, in peripheral blood B cells from untreated and nintedanib-treated IPF patients, as well as the connection with circulating BAFF.

## 2. Materials and Methods

### 2.1. Patient Characteristics and Study Design

IPF was multidisciplinarily diagnosed and managed based on current ALAT/ATS/ERS/JRS guidelines [27,28]. Cohort 1 consisted of 16 treatment-naïve IPF patients (Table 1), who were compared with 14 HCs (age: 67.1 ± 5.2; m/f: 6/8; for flow cytometry analysis of circulating B cells), and of which 12 patients were compared with 7 HCs (age: 54.2 ± 4.8; m/f: 3/4; for analysis of circulating cytokines in plasma). Peripheral blood samples from these treatment-naïve patients were collected at the time of diagnosis. In a separate cohort (cohort 2), consisting of 12 IPF patients who started nintedanib treatment, peripheral blood samples were collected at baseline (in this case, not at the time of diagnosis) and at follow-up (~1.7 ± 1.0 months) (Table 1). Ten of these twelve patients had a history of therapy prior to the start of nintedanib treatment. Sixteen HCs (age: 59.8 ± 8.0; m/f: 10/6) were used as a control group for cohort 2. HCs were selected as individuals without a history of any systemic, immunological, or pulmonary condition.

Peripheral blood was drawn by venous phlebotomy and collected in EDTA microtubes (BD Biosciences, San Jose, CA, USA) after informed consent. Peripheral blood mononuclear cells (PBMCs) and plasma were isolated using Ficoll-Paque™ (GE Healthcare, Chicago, IL, USA) density gradient centrifugation and stored at −196 °C and −80 °C, respectively, according to standard protocols. This study was approved by the medical ethics committee of the Erasmus Medical Center Rotterdam (METC 2012/512) and complies with the Declaration of Helsinki.

### 2.2. Flow Cytometric Analysis

#### 2.2.1. Flow Cytometry for B Cell Surface Markers and Intracellular BTK

Frozen PBMCs were thawed and resuspended in RPMI 1640 medium (Gibco, Thermo Fisher Scientific, Waltham, MA, USA) containing 5% fetal calf serum (Gibco; referred to as medium). A total of 5.0 × 10^5^ cells were plated in 96-well round-bottom plates and washed with MACS buffer (0.5% BSA, 2 mM EDTA in PBS). Binding of antibodies to Fc-receptors was prevented by prior incubation with TruStain FcX (Biolegend, San Diego, CA, USA) for 5 min at 4 °C. Cells were subsequently stained for surface markers with a combination of conjugated antibodies in MACS buffer for 60 min at 4 °C. After washing, cells were stained with Fixable Viability Dye (Invitrogen, Thermo Fisher Scientific) as live/dead marker, and fluorescently labeled streptavidin for 20 min at 4 °C to detect biotinylated antibodies. Cells were fixed and permeabilized with 2% paraformaldehyde and 0.5% saponin (Sigma-Aldrich, St. Louis, MO, USA), respectively, or using Cytofix/Cytoperm (BD Biosciences). Subsequent intracellular BTK staining was performed for 60 min at 4 °C, as previously described [22]. The complete staining panel is provided in Appendix A. Cells were resuspended in MACS buffer and measured on an LSR II flow cytometer (BD Biosciences).

#### 2.2.2. BCR Signaling Measurement by Phosphoflow Cytometry

Analysis of phosphorylated signaling proteins was performed by intracellular phosphoflow cytometry (phosphoflow) using an optimized protocol that allows for analysis on cryopreserved materials, essentially as described previously [29,30]. For both cohorts, samples from IPF patients and HCs were measured in single experiments to prevent inter-experimental variation. Briefly, PBMCs were thawed in medium; 3.0 × 10^5^ cells were plated per well in a 96-well round bottom plate and were brought to 37 °C. Fixable Viability Dye (Invitrogen) was added 10 min before fixation as a live/dead marker. Cells were either stimulated with 20 µg/mL goat F(ab’)_2_ anti-human Ig (α-Ig) (Southern Biotech, Birmingham, AL, USA) in medium or left unstimulated by adding plain medium. For optimal detection of phosphorylation, cells were stimulated either for 5 min (phosphorylated (p)BTK and pPLCγ2), for 2 min (pSYK), or for 1 min (pPI3K p85) at 37 °C. The stimulation was terminated by fixation with the eBioscience™ Foxp3/Transcription Factor Staining Fixation/Permeabilization Buffer (Invitrogen) for 10 min at 37 °C. Cells were then stained intracellularly for markers to define B cell subpopulations, for 30 min at 4 °C in Permeabilization Buffer (Invitrogen). pSYK Y348-PE, pBTK Y223-AF647, and pPLCγ2 Y759-AF647 were stained in Permeabilization Buffer for 30 min at room temperature (RT). pPI3K p85 Y458 was stained in Permeabilization Buffer for 15 min at RT, after which cells were washed and incubated with donkey anti-rabbit-PE (Jackson ImmunoResearch, Ely, UK; for complete staining panel, see Appendix A). Cells were resuspended in MACS buffer and measured on an LSR II flow cytometer. Flow cytometric data were analyzed with FlowJo v10 (BD Biosciences), and the geometric mean fluorescent intensity (gMFI) was determined for quantification of signal intensities.

### 2.3. Cytokine Detection by Enzyme-Linked Immunosorbent Assay (ELISA)

ELISA was performed to determine plasma levels of BAFF (R&D Systems, Minneapolis, MN, USA) and transforming growth factor (TGF)-β1 (Invitrogen) according to the manufacturer’s instructions. Optical density (OD) values were measured at 450 nm and subtracted from 570 nm background, using a SpectraMax plate reader (Molecular Devices, San Jose, CA, USA). Samples were measured in duplicate, blanked against untreated wells, and concentrations were calculated by averaging the duplicate OD values and subsequent interpolation from the standard curve.

### 2.4. Statistical and Computational Analysis

Graphs were created and statistics were calculated using GraphPad Prism 9 software (GraphPad Prism Inc., San Diego, CA, USA). A Shapiro–Wilk test was performed to test for Gaussian distribution within each group. If the compared groups showed a Gaussian distribution, an unpaired or paired Student’s *t*-test was performed. In the case of a nonparametric distribution, a Mann–Whitney *U* test was performed. For correlation analysis, a Pearson correlation coefficient (*r*; in the case of a normal distribution) or a Spearman’s rank correlation coefficient (ρ; in the case of a nonparametric distribution) was calculated. Survival Kaplan–Meier curves were analyzed using a log rank test for statistical significance. A *p*-value < 0.05 was considered statistically significant.

Principal component (PC) analysis (PCA) was performed using R version 3.6.1 and RStudio version 1.2.5001. PCA plots were built and the relative contribution of each variable to the PCs was calculated using the *FactoMineR* and *factoextra* package [31].

## 3. Results

### 3.1. Naïve B Cells from IPF Patients Show Aberrant BCR Signaling upon Stimulation

To study BCR responsiveness, total PBMCs from HCs and treatment-naïve IPF patients were left unstimulated or were stimulated with α-Ig. Using phosphoflow cytometry, the phosphorylation status of several signaling molecules was determined in different B cell subsets (gating strategy in Appendix A). In these experiments, B cells from both IPF patients and HCs showed a robust response to BCR stimulation, as for all phosphorylated proteins tested, we found a significant difference in signal between unstimulated and α-Ig stimulated B cells (data not shown). Given the previously found increased BTK expression in B cells in a major fraction of IPF patients [18], we first determined BTK phosphorylation status. In unstimulated B cells, pBTK (Y223) levels were similar between HCs and IPF patients (Figure 1A). However, following stimulation of the BCR, B cells from IPF patients showed enhanced BTK phosphorylation compared with HCs. When this was analyzed separately for different B cell subpopulations, only α-Ig-stimulated naïve—and not memory—B cells showed increased BTK phosphorylation compared with α-Ig-stimulated naïve and memory B cells from HCs (Figure 1A). Notably, the proportion of naïve B cells unresponsive to BCR stimulation, as inferred by the pBTK-low population following α-Ig stimulation, was significantly lower in IPF patients than in HCs (~20.2% versus ~40.7%, respectively) (Figure 1A). Within the pBTK-high population, the signal intensity in naïve B cells was also greater in IPF patients than in HCs (Appendix A). The enhanced pBTK signal in α-Ig-stimulated naïve B cells from IPF patients, as compared with HCs, can therefore be attributed to both a substantial reduction in the fraction of unresponsive B cells and enhanced phosphorylation within the responding naïve B cell population.

Next, phosphorylation of BCR signaling molecules upstream of BTK, SYK (Y348), and directly downstream of BTK, PLCγ2 (Y759), was measured. pSYK (Figure 1B) and pPLCγ2 (Figure 1C) levels were comparable in unstimulated B cells from IPF patients and HCs. Like BTK, phosphorylation of SYK and PLCγ2 was increased in naïve but not in memory IPF B cells compared with HCs upon α-Ig stimulation, although significance was only reached for pPLCγ2. The proportion of pPLCγ2-low naïve B cells after stimulation was lower in IPF patients than in HCs (~26.2% versus ~41.9%, respectively) (Figure 1C). When selecting for pSYK- and pPLCγ2-high naïve B cells, no difference was found in signal intensity between IPF patients and HCs (Appendix A). Thus, for pSYK and pPLCγ2, the decreased fraction of unresponsive naïve B cells was responsible for the increase in overall phosphorylation signals in IPF patients. 

The enhanced BCR responsiveness was not due to an increased expression of BCRs, as surface IgD and IgM expression was similar between HC and IPF naïve B cells (Appendix A). No differences were found between IPF patients and HCs in α-Ig-induced phosphorylation of BTK, SYK, or PLCγ2 in the individual memory B cell subsets, such as IgD^+^, IgM^+^, or class-switched cells (data not shown).

Together, these data indicate that, similar to patients with a systemic autoimmune disease [21], naïve B cells from IPF patients display aberrant BCR signaling following stimulation.

### 3.2. Enhanced BCR Signaling in Naïve B Cells from IPF Patients Correlates with TACI Expression

To assess whether the increase in BCR signaling found in naïve B cells from IPF patients was associated with an altered B cell phenotype, we measured intracellular BTK protein, surface activation marker CD86, BAFFR, and transmembrane activator and calcium-modulator and cyclophilin ligand interactor (TACI) expression in different B cell subsets (gating strategy in Appendix A). We did not detect an altered expression of these markers in IPF patient compared with HC B cells (Appendix A). Principal component analysis (PCA) of the BCR signaling molecules and these B cell markers measured in naïve B cells, however, indicated that IPF patients and HCs significantly separated on principal component 1 (PC1) (Figure 2A). This component predominantly comprised the phosphorylation status of BCR signaling molecules and the associated proportions of phosphorylation-low B cells following stimulation (Figure 2B). Hereby, the contribution of signaling molecule phosphorylation of unstimulated and α-Ig-stimulated naïve B cells to PC1 appeared to be similar. Expression levels of BTK and CD86, the most important contributors to PC2, did not significantly differentiate between HCs and IPF patients.

Next, we created a correlation matrix for the phosphorylation status of BCR signaling molecules measured in vitro in unstimulated and stimulated conditions, and the expression levels of B cell surface and intracellular markers ex vivo. We found that in HCs and IPF patients, phosphorylation of the signaling molecules within the BCR signalosome (BTK, SYK, and PLCγ2) strongly correlated with one another, in unstimulated conditions and following BCR stimulation in naïve B cells (Figure 2C,D), as well as in unstimulated memory B cells (Appendix A). For both IPF patients and HCs, the correlation between pBTK and pPLCγ2 is shown as an example (Figure 2E,F). A positive correlation was found for unstimulated and stimulated naïve B cells from HCs between BCR signalosome phosphorylation and ex vivo BAFFR surface expression levels (Figure 2C,E; pPLCγ2 used as an example). This reached significance for pSYK and pPLCγ2 in stimulated cells. In contrast, in both unstimulated and stimulated naïve B cells from IPF patients, this correlation between BCR signalosome phosphorylation and ex vivo BAFFR surface levels was not found but had apparently shifted towards TACI surface expression (Figure 2D,F; pPLCγ2 used as an example). In the memory B cell compartment of HCs and IPF patients, we observed a similar phenomenon (Appendix A). Despite the absence of detectable differences in the mean BTK protein expression levels in B cells from HCs and the IPF patients within this cohort (Appendix A), the BCR signalosome of unstimulated IPF B cells strongly correlated with BTK expression, while this correlation was absent in HCs (Figure 2C,D).

Taken together, these data show that in naïve B cells from IPF patients, BCR signaling is dysregulated and, in contrast to HCs, does not correlate with BAFFR but instead with TACI surface expression levels.

### 3.3. Negative Correlations between Circulating BAFF and BAFFR Expression, and between Circulating TGF-β and Phosphorylation of BCR Signalosome Molecules Following Stimulation

As a pleiotropic cytokine, TGF-β plays an important role in stimulating fibrosis, though it is also known for its regulatory effects on lymphocytes [32]. We observed a trend towards increased circulating levels of TGF-β in IPF patients compared with HCs (Figure 3A). Circulating TGF-β levels showed a negative correlation with phosphorylation of BTK following BCR stimulation, specifically in naïve B cells (Appendix A). The same negative trend was observed for the other BCR signalosome molecules, SYK and PLCγ2, but these were not significant (data not shown).

As a ligand for BAFFR and TACI, BAFF plays a crucial rule in B cell survival and thus in peripheral selection of autoreactive B cells and the maintenance of the mature B cell pool [33]. Because the BAFF–BAFFR axis and BCR signaling are intertwined and positively regulate one another [25,34], we hypothesized that in IPF patients, BAFFR and TACI expression on B cells and the altered BCR signaling were associated with increased circulating BAFF levels. In line with reported findings [12], circulating BAFF levels were significantly increased in IPF patients compared with HCs (Figure 3B). A negative correlation was observed between circulating BAFF levels and BAFFR surface expression on B cells, which reached significance for the memory B cell compartment (Figure 3C). We could not detect significant correlations between circulating BAFF levels and BTK expression, TACI expression, or phosphorylation of BCR signaling molecules in B cells from IPF patients (Appendix A, and data not shown).

We observed reduced survival in patients with high BAFF levels, compared with patients with low BAFF levels, as previously reported [12] (Appendix A). Conversely, patients with low BAFFR expression on naïve B cells tended to have reduced survival compared with patients with high BAFFR expression (Appendix A). No association was observed between BTK expression, TACI expression, or the phosphorylation of the BCR signalosome molecules, and survival (data not shown).

In summary, our data indicate that in IPF patients, increased circulating BAFF levels correlate with reduced BAFFR expression and that increased circulating TGF-β levels are associated with decreased BCR signaling.

### 3.4. Nintedanib Treatment Induces Substantial Changes in BCR Signaling in Naïve and Memory B Cells

Nintedanib is a receptor tyrosine kinase inhibitor which suppresses downstream signaling of FGFR and VEGFR, thereby dampening pro-fibrotic processes in IPF patients [35]. However, non-receptor tyrosine kinases are also predicted targets of nintedanib, such as the Src family of kinases including lymphocyte-specific protein tyrosine kinase (LCK), tyrosine-protein kinase (LYN), and proto-oncogene c-Src (SRC) [36]. More recently, BTK was also added to that list of predicted targets [37]. Because these molecules are crucial in BCR signaling, we hypothesized that BCR signaling in B cells from IPF patients may be affected by nintedanib treatment. To that end, we investigated BCR signaling in a second cohort of IPF patients. We analyzed peripheral blood samples from patients just before and on average 1.7 months after the start of nintedanib treatment. In this cohort, the same BCR signalosome molecules were investigated as in the first cohort, with the addition of PI3K p85 (Y458). We found that after the start of nintedanib treatment, mean pBTK levels significantly increased in α-Ig-stimulated naïve B cells—but not in α-Ig-stimulated memory B cells—from IPF patients (Figure 4A). The average levels of upstream pSYK and pPI3K p85, and downstream pPLCγ2 in both unstimulated and α-Ig stimulated conditions were not significantly altered following treatment (Figure 4B–D). When these IPF patients, analyzed before and after the start of treatment, were compared with a HC cohort, no significant differences in the phosphorylation of these four signaling molecules were observed (data not shown).

Interestingly, however, a considerable trend was observed between paired samples following α-Ig stimulation: patients that showed high phosphorylation of the BCR signalosome molecules before treatment, showed low phosphorylation after the start of nintedanib treatment, and vice versa. This phenomenon was observed in both naïve and memory B cells and was reflected by strong negative correlations when comparing B cells from patients before and after the start of nintedanib treatment (shown for pSYK and pPLCγ2 in naïve B cells in Figure 4E). However, pBTK did not follow this pattern (Figure 4F).

Taken together, these results indicate that in IPF patients, BCR signaling in peripheral blood naïve and memory B cells is markedly affected by nintedanib treatment.

## 4. Discussion

Evidence has been accumulating that B cells and autoimmunity are involved in the pathogenesis of IPF. In this study, we used phosphoflow cytometry to distinguish different B cell subsets and to simultaneously analyze phosphorylation of several critical downstream BCR signaling molecules in peripheral blood B cells from two cohorts of IPF patients and healthy controls. We found that naïve B cells, but not memory B cells, from treatment-naïve IPF patients displayed increased phosphorylation of the BCR signaling molecules SYK, BTK, and PLCγ2 following BCR stimulation in vitro. This enhanced BCR signaling correlated with surface expression levels of TACI but not with BAFFR, both of which bind BAFF and therefore provide pro-survival signals to B cells. Remarkably, treatment of IPF patients with the anti-fibrotic tyrosine kinase inhibitor nintedanib induced major changes in BCR signaling. Taken together, our findings do not only support the involvement of B cells in the pathogenesis of IPF, but also suggest that targeting BCR signaling might contribute to the therapeutic effect of nintedanib.

Developing B cells undergo selection on the basis of BCR reactivity and signaling strength, preventing the generation and activation of autoreactive B cells [33,38]. Within the circulating naïve B cell pool, autoreactive naïve B cells are kept in check by a decreased or even lack of response to BCR stimulation, referred to as B cell anergy [39]. In this context, it is notable that our phosphoflow analysis showed that the enhanced overall phosphorylation signals following BCR stimulation in the naïve B cell population from IPF patients was due to a reduced percentage of naïve B cells unresponsive to BCR stimulation. Therefore, it is tempting to speculate that the induction or maintenance of an anergic state in a—possibly autoreactive—fraction of naïve B cells is defective in IPF patients. This would also be in line with the absence of aberrant BCR signaling within the memory B cell population of IPF patients.

BAFF plays an important role during B cell development and the negative selection of autoreactive B cells, as increased levels can lead to dysfunctional negative selection and systemic autoimmunity [40,41,42]. Since IPF is associated with repetitive alveolar epithelial injury, it is conceivable that locally infiltrated B cells in close contact with local debris gain access to autoantigens. Increased BAFF levels could help to overcome negative selection of activated B cells in the absence of T cell help [33,43,44]. Indeed, studies using animal models have shown the importance of BAFF in pulmonary fibrosis [43,44]. Next to the BAFFR, BAFF can bind to TACI, and evidence has been provided that enhanced BAFF levels, at least partially, exert their pathogenicity in autoimmunity through TACI [45,46]. Though incompletely understood, TACI has both stimulatory and regulatory roles in B cell responses, and a balanced expression is crucial [47]. Because we observed that phosphorylation of BCR signaling molecules in naïve B cells correlated in HCs with BAFFR expression, but in IPF patients with TACI expression, defective fine-tuning of BAFF responses may contribute to immunopathology in IPF.

TGF-β is well-known for its pathogenic pro-fibrotic properties, though it also functions as an anti-inflammatory cytokine. We observed a trend towards increased TGF-β levels in the circulation of IPF patients, which showed a negative correlation with α-Ig-induced phosphorylation of BCR signalosome molecules. This is in line with previously reported suppressive effects of TGF-β on BCR signaling and immunoglobulin production in vitro [48,49]. However, it remains unclear how the observed correlation connects to disease severity or stage, because increased levels in circulation are not consistently found [50,51,52,53]. IPF therapies are largely focused on specific inhibition of pathogenic TGF-β signaling [54]. Because of the known immunoregulatory effects of TGF-β, including induction of regulatory T cells [55], it is important to investigate whether inhibition of TGF-β may be accompanied by increased BCR signaling or inflammatory responses.

We detected alterations in BCR signaling after the start of nintedanib treatment, as BCR signaling characteristics of patient B cells seemed inverted. Nintedanib is thought to suppress pro-fibrotic pathways by inhibiting VEGFR, FGFR, and platelet-derived growth factor receptor signaling. However, various molecules involved in BCR signaling, including LYN, SRC, and BTK, are also predicted targets of nintedanib with expected IC_50_ within the physiological treatment range [36,37,56]. Moreover, nintedanib has the capacity to activate SH2 domain-containing phosphatase-1 (SHP-1), a negative regulator of BCR signaling [57]. Therefore, our findings could be driven by B cell intrinsic effects of nintedanib. Alternatively, nintedanib treatment may affect other signaling pathways in other cell types such as T cells or affect the cytokine milieu, which would subsequently affect BCR signaling [58,59,60,61]. Irrespective of the mechanisms involved, it can be concluded that part of the beneficial effects of nintedanib treatment might be due to inhibitory effects on signaling pathways in lymphocytes.

It is currently disputed whether B cell-targeting therapies are beneficial for IPF patients [15,16,62], and a large trial with rituximab in CTD-ILD is currently ongoing (NCT01862926) [63]. In mouse models, BTK inhibition shows divergent effects [64,65]. Nevertheless, clinical studies targeting B cells in IPF are currently ongoing or have recently been performed (NCT03287414, NCT01969409), and BTK targeting in IPF is still of interest [65].

Our study has some limitations. In both cohorts, we analyzed small numbers of patients. Since IPF is a heterogenous disease, our study did not allow for the identification of correlations with clinical parameters or survival of these patients, which would require larger numbers of treatment-naïve patients. Ideally, future studies should include functional analyses of BAFF receptor and TACI signaling. Furthermore, possibly due to the low number of patients analyzed, we were not able to replicate our previous findings of increased BTK levels in B cells in a subset of IPF patients. Nevertheless, clear differences in phosphorylation of BTK, its upstream kinase SYK, and its downstream substrate PLCγ2 were observed following BCR stimulation. Our assay on BTK phosphorylation, therefore, seems more sensitive and physiologically more relevant, as it reflects BTK enzymatic activity instead of only protein levels. In addition, we were able to identify remarkable correlations, e.g., between the phosphorylation of BCR signaling molecules, and surface BAFFR and TACI expression, as well as the effects of nintedanib on BCR signaling.

Therefore, our results contribute to the evidence for a role for B cells in IPF pathogenesis and, most interestingly, point to abnormalities in naïve B cells. Further studies building on these findings and involving larger patient cohorts should clarify whether targeting BCR signaling, for example with currently available specific small molecule inhibitors [66], has potential value as a treatment option for IPF patients.

## Figures and Tables

**Figure 1 cells-10-01321-f001:**
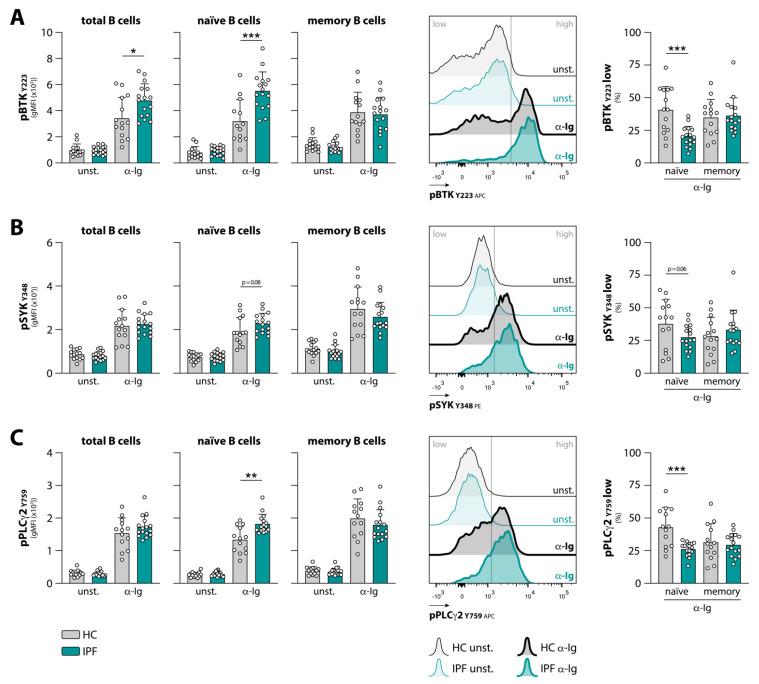
Phosphoflow analysis of BCR downstream signaling molecules in peripheral blood B cells from HCs and IPF patients (cohort 1). (**A**–**C**) Phosphorylation levels, expressed as geometric mean fluorescence intensity (gMFI) under unstimulated (unst.) conditions and after BCR stimulation (anti-immunoglobulin; α-Ig) in total (CD3^−^CD19^+^IgM^+/−^), naïve (CD3^−^CD19^+^CD38^lo^CD27^−^IgD^+^; 56.8% ± 15.4 in HCs and 69.0% ± 11.0 in IPF patients of total B cells), and memory (CD3^−^CD19^+^CD38^lo^CD27^+^; 23.6% ± 5.8 in HCs and 21.7% ± 6.2 in IPF patients of total B cells) B cells from HCs and IPF patients for (**A**) pBTK _Y223_, (**B**) pSYK _Y348_, and (**C**) pPLCγ2 _Y759_ (*left panel*). Representative histograms are shown with the gray vertical line indicating the gate for selecting the population low/high in phosphorylation following α-Ig stimulation (*middle panel*). Proportions of (**A**) pBTK-low, (**B**) pSYK-low, and (**C**) pPLCγ2-low cells, expressed as percentage of total naïve or memory B cell fraction (*right panel*). Subjects are indicated by individual data points, and bars indicate mean values + SD. * *p* < 0.05, ** *p* < 0.01, *** *p* < 0.001 in ((**A**–**C**) (right panel)) by an unpaired two-tailed Student’s *t*-test or (in (**C**) (left panel)) by a Mann–Whitney *U*.

**Figure 2 cells-10-01321-f002:**
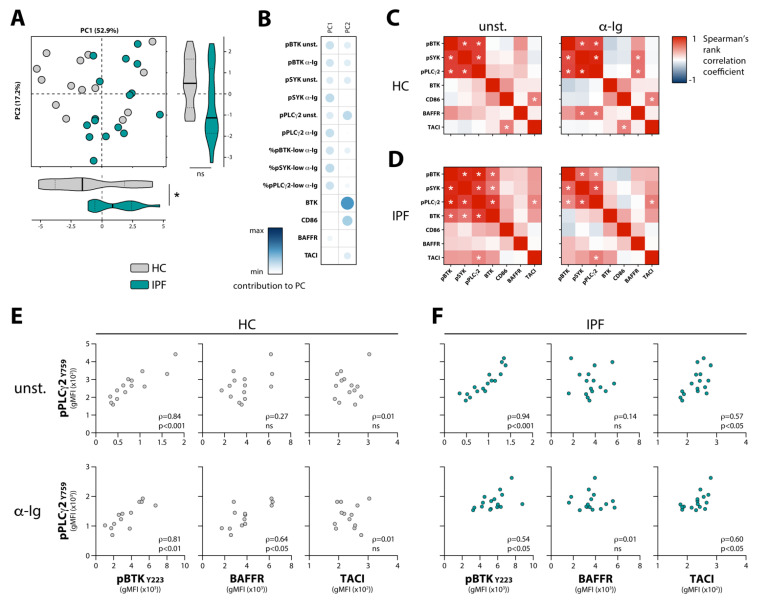
Correlation analysis of ex vivo measured B cell markers and in vitro measured phosphorylation of BCR signalosome molecules in naïve (CD3^−^CD19^+^CD38^lo^CD27^−^IgD^+^) B cells from HCs and IPF patients (cohort 1). (**A**) Principal component analysis (PCA) of BCR signalosome molecules and B cell activation markers and (**B**) the individual contribution of each variable to PC1 and PC2 from IPF and HC naïve B cells. (**C**,**D**) Spearman’s rank correlation matrix for (**C**) HCs and (**D**) IPF patients for the indicated markers measured ex vivo and phosphorylated proteins of unstimulated (unst.) and BCR-stimulated (α-Ig) naïve B cells. (**E**,**F**) Spearman’s rank correlation analysis of (**E**) HCs and (**F**) IPF patients for the correlation of pPLCγ2 with pBTK, BAFFR, and TACI in unstimulated (unst.) and BCR-stimulated (α-Ig) naïve B cells. Subjects are indicated by individual data points. * *p* < 0.05 in (**A**) by an unpaired two-tailed Student’s *t*-test; ns, not significant.

**Figure 3 cells-10-01321-f003:**
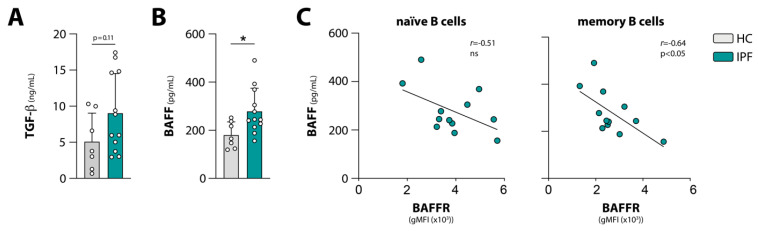
Analysis of circulating BAFF and TGF-β levels in HC and IPF patients (subset of cohort 1). (**A**) TGF-β levels in plasma of HCs and IPF patients measured by ELISA. (**B**) BAFF levels in plasma of HCs and IPF patients measured by ELISA. (**C**) Pearson correlation analysis between BAFF levels in plasma and BAFFR surface expression on naïve (CD3^−^CD19^+^CD38^lo^CD27^−^IgD^+^) and memory (CD3^−^CD19^+^CD38^lo^CD27^+^) B cells in IPF patients. Subjects are indicated by individual data points, and bars indicate mean values + SD. * *p* < 0.05 in (**B**) by an unpaired two-tailed Student’s *t*-test.

**Figure 4 cells-10-01321-f004:**
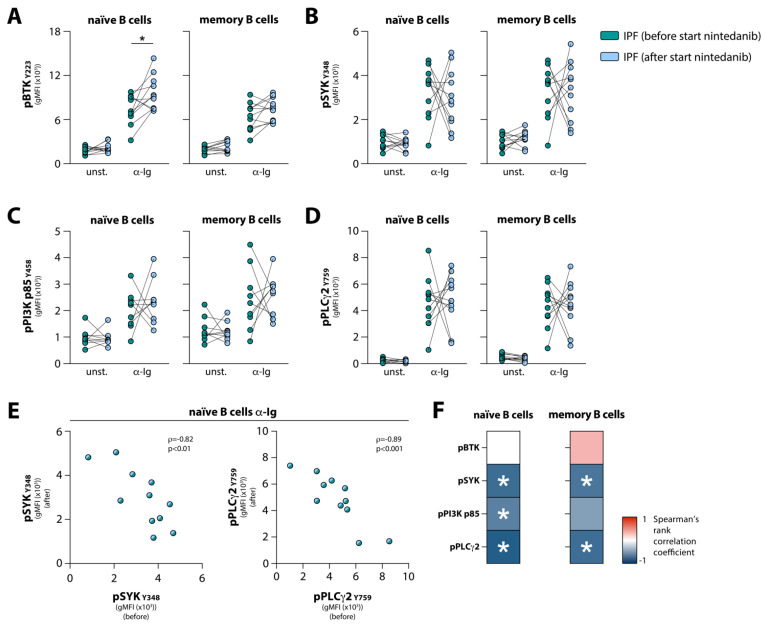
Phosphoflow analysis of BCR downstream signaling in nintedanib-treated IPF patients’ peripheral blood B cells (cohort 2). (**A**–**D**) Phosphorylation levels in unstimulated (unst.) conditions and after α-Ig stimulation for (**A**) pBTK _Y223_, (**B**) pSYK _Y348_, (**C**) pPI3K p85 _Y458_, and (**D**) pPLCγ2 _Y759_ in naïve (CD3^−^CD19^+^CD38^lo^CD27^−^IgD^+^) and memory (CD3^−^CD19^+^ CD38^lo^CD27^+^) B cells from IPF patients before and after the start of nintedanib treatment. (**E**,**F**) Spearman’s rank correlation analysis of the phosphorylation levels for (**E**) pSYK (*left*) and pPLCγ2 (*right*) in naïve B cells following BCR stimulation, before versus after the start of nintedanib treatment. (**F**) Spearman’s rank correlation matrix for phosphorylation of BCR signalosome molecules before and after the start of nintedanib treatment. Subjects are indicated by individual data points. * *p* < 0.05 in (**A**) by a paired two-tailed Student’s *t*-test.

**Table 1 cells-10-01321-t001:** IPF patient characteristics.

	IPF Cohort 1	IPF Cohort 2
subjects, n	16	12
age (years), mean (±SD)	68.5 (±7.2)	69.1 (±7.7)
male sex, n	10 (63%)	9 (75%)
time to diagnosis (years), mean (min–max)	3.1 (0.1–12.0)	1.8 (0.2–7.0)
CT diagnosis, definite UIP/probable UIP/inconsistent *	8/6/2	5/6/1
smoking, active/previous/never	0/15/1	1/11/0
PY (years), mean (min–max) *	23.0 (0–80)	24.8 (1–60)
FVC (L), mean (±SD) *	2.8 (±0.8)	2.5 (±1.0)
FVC % predicted, mean (±SD)	80.1 (±14.1)	75.3 (±22.9)
Tiffeneau index, mean (±SD)	82.9 (±5.4)	77.3 (±11.2)
prednisone, n	0	0
nintedanib, n	0	0
pirfenidone, n	0	10 (83%)
time between last intake of medication and starting nintedanib (weeks), mean (±SD)	-	2.3 (±2.1)

* CT, computed tomography; UIP, usual interstitial pneumonia; PY, pack years; FVC, forced vital capacity.

## Data Availability

The data presented in this study are available on request from the corresponding authors.

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
