# Peer review of "Aberrant B Cell Receptor Signaling in Naïve B Cells from Patients with Idiopathic Pulmonary Fibrosis"

_cells, 2021, doi:10.3390/cells10061321_

Round 1
Reviewer 1 Report
This is an interesting and relatively novel study, that may have considerable significance. The author's methods (with a possibly important exception), data analyses and presentation (albeit complexities of the latter) and writing are generally very good. I do have concerns, however, that detract from the potential value of this report:
1) The numbers are small, especially considering the number of assays/measures and interactive complexities of some analyses, and perhaps questionable diagnoses of some subjects (Table 1).
2) Is cohort 2 a subpopulation of cohort 1? Their relationship should be more clearly identified.
3) Why were these studies performed on previously frozen PBMNC? Do we know that cryopreservation does not alter relative proportions of B-cell subpopulations or, and much more likely, alter functions of these cells? I have found short-term freezing alters proportions of lymphocytes and alters many effector functions of lymphocytes among the survivors (and it gets worse the longer they were stored) and am loathe to use these in lieu of "fresh cells". Were both populations (HC and IPF) stored the same duration. At the least, the authors need to validate their assays/measures are unaffected by these preparations. And more details would be nice too: duration, comparability among populations, methods, percent recovery, etc.
4) Most flow assays here are based on MFI determinations. Reproducibility of MFI measures over time, particularly in comparisons of distinct populations, can be problematic, even when the cytometer is appropriately calibrated. Were these assays all run at the same time or separated in time? How were these standardized?
5) Differences of B-cell phenotypes and subpopulation numbers/proportions among IPF and HC have been reported - what was found here (this may have some relevance too for concerns about cryopreservation).
6) The histograms in Fig 1 seem to be "adjusted" for height (number of cells), given their similarities despite different MFI intensity differences. If so, this manipulation is misleading. Regression lines are not appropriate in Spearman's correlations.
7) Aside from potential issues of methodologies/preservation artifacts, etc., the repeated contention here that Ofev "markedly affects" BCR signaling may arguably be an overinterpretation. The drug did seem to alter pBTK, but had not seeming effect on pPLCg2 or upstream signaling. No other effector functions (e.g., activation, differentiation, or Ig production) were measured, nor were effects on HC B-cells. Since BTK activation could be at least in part due to pathways other than the BCR, even if this observation is real, there could be another explanation(s).
8) The absence of other functional studies (e.g., addition of BAFF or blocking BAFF, etc.) is a limitation.
9) The Discussion could be shortened significantly.
Reviewer 2 Report
This is an interesting paper which aims to increase our knowledge on the presence and potential role of specific B cell populations in IPF. While this is generally a well written and well-presented paper. I do have some concerns that need to be addressed. At the end of the paper I am left wondering what the significance of the B cell response to the progression and management of IPF is. How can we use this information, what does this mean for patient treatment? Another main concern is the amount of data that has been represented in the manuscript as “data not shown”. In my opinion if the data is important enough to mention it should be included either in the body of the manuscript or in the supplementary data. I also have a few specific points/queries which I have listed below for your considerations;
1. In the methods section 7 HC are listed as being used for cohort 1 and 16 for cohort 2. While demographic information has been provided for both the IPF cohorts no information is provided on the HC, are these samples age-matched? Can you please provide all relevant information. This is critical because as we know B cell phenotypes change with age. Also it would be useful to know if any of the controls (if aged matched) have any underlying conditions such as rheumatoid arthritis etc.
2. Can you please clarify which cohort has been used for which analysis (this information should be included in each of the figure legends. I presume Cohort 2 was used for Fig 1 as the n numbers for HC are >7.
3. % gMFI is graphed. It would also be useful to know the % of B cells expressing each of these proteins and what % of the patient samples were memory vs naïve B cells.
4. In Figure 1S can you please provide the FSC/SSC and the viability dye information. Can you please also indicate the number of events counted and the % of total viable cells within each gate.
5. Figure 1. There is no statistical value for the comparison of untreated vs Igα-treated cells for PLCy2. I also wonder if the statement line 191 “Thus, also for pPLCy2……” needs to be preceeded by a sentence to state what the data shows and then qualified by thus this data suggests…
6. Can you please state which cohort has been used and confirm the n# per group in the figure legend for Figure 2. It looks like less that the n number stated for cohort 2. Have any donors been removed from the analysis? Can you please include a graph to demonstrated the increased expression of TACI in IPF vs HCs. Functional analysis to confirm that TACI surface expression is required for BCR signalling would support your concluding statement.
7. In the concluding statement for the data presented in Fig2 the authors state “Taken together,………..,does not correlate with BAFFR but instead with TACI surface expression” but in Figure 3 the authors focus on the relationship between BAFF and the BAFFR. The data (even though negative, on TACI should be included rather been cited as data not shown. Do you see a similar relationship with other ligands such as APRIL?
8. You identified a negative correlation between TGFß serum levels and BTK activation following BCR stimulation. Can you clarify what this means in terms of disease state.
9. The authors state that increased TGFß levels are associated with decreased BCR signalling. It would be useful to know if BCR signalling is altered in individuals treated with pirfenidone which is known to reduced TGFß synthesis. The authors show that BCR signalling is altered in individuals post nintedanib treatment but it is not clear what the mechanism is. How can BCR signalling be targeted therapeutically?
Reviewer 3 Report
This exceptionally well-written paper, entitled, “Aberrant B cell receptor signaling in naïve B cells from patients 2 with idiopathic pulmonary fibrosis”, which presents contributing evidence that support the involvement of B cells in Idiopathic pulmonary fibrosis pathogenesis and, furthermore, suggest that targeting B cell receptor signaling has potential value as treatment option. Figures within the manuscript are nicely paneled and appropriately present key findings. Also, appropriately stated are the limitations of this study in that it presents findings from a small number of patients with idiopathic pulmonary fibrosis that, of itself, is heterogeneous. Thus, many clinical correlations were not possible. Finally, that targeting B cell receptor signaling may have some potential with regards to the treatment of idiopathic pulmonary fibrosis is suggested and, based upon this research, is supported by these findings.
Author Response
We would like to thank the reviewer for his/her time and positive words.Round 2
Reviewer 1 Report
The authors have largely addressed my original concerns, at least when feasible. I am still a bit concerned about making comparisons and inferences from such very small populations, and despite contentions of the authors, I remain a little worried that the focus on cryopreserved cells may have introduced some occult biases or confounding.
I do understand the reasons for these limitations/concerns of the research methods.
Reviewer 2 Report
This reviewer is satisfied that the comments have been adequately adressed.